



# Validation of the dynamic wake meandering model with respect to loads and power production

Inga Reinwardt[1], Levin Schilling[1], Dirk Steudel[2], Nikolay Dimitrov[3], Peter Dalhoff[1], and
Michael Breuer[4]

[1]Dep. Mechanical Engineering & Production, HAW Hamburg, Berliner Tor 21, D-20099 Hamburg, Germany
[2]Dep. Turbine Load Calculation, Nordex Energy GmbH, Langenhorner Chaussee 600, D-22419 Hamburg, Germany
[3]Dep. of Wind Energy, DTU, Frederiksborgvej 399, 4000 Roskilde, Denmark
[4]Dep. of Fluid Mechanics, Helmut-Schmidt University Hamburg, Holstenhofweg 85, D-22043 Hamburg, Germany

**Correspondence:** Inga Reinwardt (inga.reinwardt@haw-hamburg.de)

**Abstract.** The outlined analysis validates the dynamic wake meandering (DWM) model based on loads and power production measured at an onshore wind farm with small turbine distances. Special focus is given to the performance of a version of the DWM model that was previously recalibrated at the site. The recalibration is based on measurements from a turbine nacelle-mounted lidar. The different versions of the DWM model are compared to the commonly used Frandsen turbulence model.

The results of the recalibrated wake model agree very well with the measurements, whereas the Frandsen model overestimates the loads drastically for short turbine distances. Furthermore, lidar measurements of the wind speed deficit as well as the wake meandering are incorporated in the DWM model definition in order to decrease the uncertainties.

## 1   Introduction

Wake models are a key aspect in every site-specific load calculation procedure. The used wake model has significant impact on
predicted loads and the power output of the whole wind farm, hence, an accurate wake model is of major importance for a wind farm design optimization process. Planning a new wind farm is a highly iterative process, where time-consuming calculations are avoided as far as possible, so that the complexity and the accuracy of the model need to be well balanced.

Simple analytical wake models can be divided into models estimating either the mean wind speed reduction in the wake or the wake-induced turbulence. While the former serves as a basis for power calculations, the latter is necessary to compute
loads. One of the main simple analytical models for calculating the wake-induced turbulence in a wind farm is the so-called Frandsen model (see, e.g., Frandsen (2007)). Reinwardt et al. (2018) and Gerke et al. (2018) have shown that this model delivers conservative results, especially for short turbine distances, a limitation that is critical for onshore wind farms in densely populated areas, where a high energy output per utilized area is crucial. Another simple, but less common, analytical model to calculate the wake-induced turbulence is introduced in Quarton and Ainslie (1989). Jensen (1983) provides an analytical
model to predict the wind speed reduction in the wake. More recently developed wind speed reduction models can be found in Larsen (2009) and Bastankhah and Porté-Agel (2014). The latter is based on a Gaussian distribution for the velocity deficit in the wake. A more sophisticated model for calculating the wind speed deficit expansion in the wake is explained in Ainslie



(1988), where the author suggests to solve the thin shear layer approximation of the Navier-Stokes equations with an eddy viscosity closure approach.

The Dynamic Wake Meandering (DWM) model investigated here is strongly influenced by the work of Ainslie (1988). It describes the physical behavior of the wake more precisely, while it is still less time-consuming and complex than a complete computational fluid dynamic (CFD) simulation. Moreover, it is capable of estimating the wake-induced turbulence as well as the wind speed deficit. The model assumes that the wake behaves like a passive tracer, i.e., the wake itself moves in vertical and horizontal direction (Larsen et al., 2008b). The meandering motion in combination with the shape of the wind speed deficit in

the meandering frame of reference (MFR) lead to increased turbulence at the wake-affected turbine and thus plays an eminent role for the loads of the downstream turbine. As of late, the DWM model is included in the new edition of the IEC guideline (IEC 61400-1 Ed.4). It was validated and calibrated with actuator disk and actuator line simulations as outlined in Madsen et al. (2010), whereas a validation of the model with measured loads and power production was carried out in Larsen et al. (2013). Keck (2015) presents a power deficit validation of a slightly different version and extension of the model towards a standalone

implementation.

The DWM model has proved to be more accurate in load prediction than the commonly used Frandsen model (Reinwardt et al., 2018). Furthermore, Reinwardt et al. (2020) present a recalibrated version of the model, which provides a very precise description of the wind speed deficit in the MFR. The authors investigate the impact of the ambient turbulence intensity (TI) on the eddy viscosity definition in the description of the wind speed deficit in the MFR based on lidar measurements from a wind

farm to determine an improved correlation function. The same wind farm is used in the present study. In the following analysis, the recently calibrated version of the DWM model is validated with respect to loads and power production and compared to the original model definition. A further analysis of the recalibrated model beyond the wake wind deficit is necessary to investigate the influence of the recalibration on loads and power production.

Further research on wake model validation and fatigue loads in wake conditions can be found in Thomsen and Sørensen

(1999) and Madsen et al. (2005). Studies related to wake model validation based on power output measurements and wind farm efficiency calculations are outlined in Barthelmie et al. (2007a), Barthelmie et al. (2007b), and Barthelmie and Jensen (2010). Furthermore, the Jensen wake model was recalibrated based on power measurements in Cleve et al. (2009) and Duc et al. (2019).

Besides the validation of the recalibrated model according to power output and loads, in the present study, lidar wake

measurements are integrated into the load simulation to support the calculation and decrease the uncertainties. The measured wind speed deficit in the MFR and the time series of the meandering are introduced successively. Related studies with a different approach of integrating the lidar measurements are Dimitrov et al. (2019) for wake-free inflow conditions, and Conti et al. (2020) for wake conditions. In comparison to the outlined methods the approach investigated here does not need any high frequency or raw data from the lidar system. It is purely based on the measured line of site (LOS) wind speed. Furthermore, the

outlined anaylsis focuses on the measured wind speed deficit and the meandering of the wake, which is successively introduced in the DWM model definition, whereas in Conti et al. (2020) special focus is given to the estimation of turbulence in the wake. The wake turbulence is only indirectly captured here by the investigated wake meandering and the wind speed deficit gradient





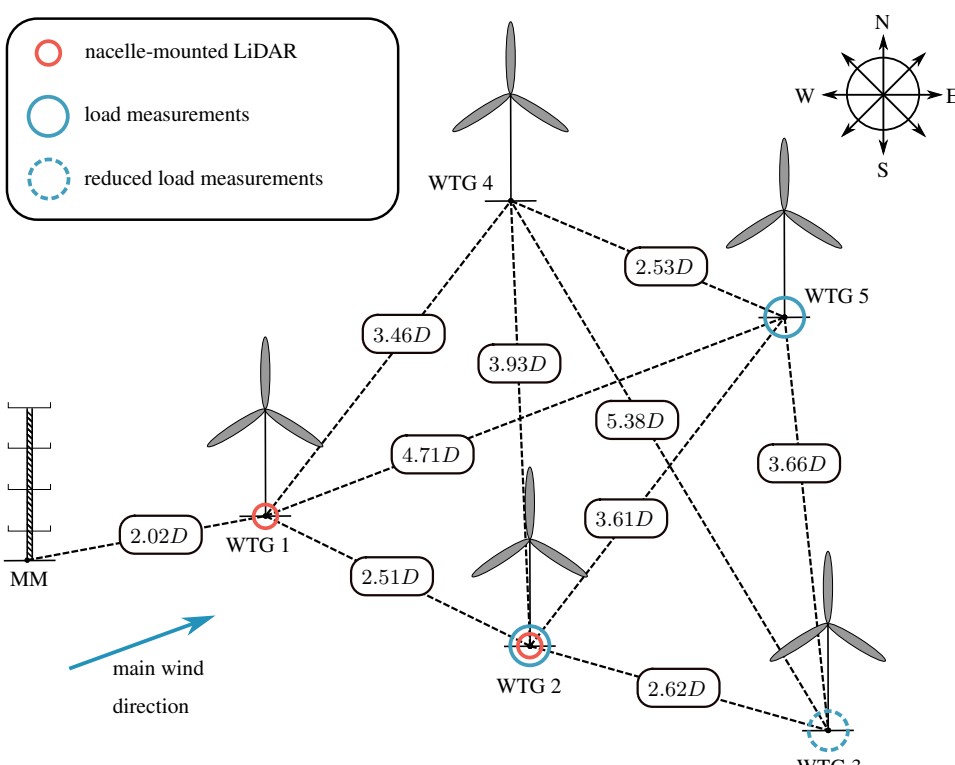

**Figure 1.** Wind farm layout with measurement equipment (Reinwardt et al., 2020).

in the MFR. The wake meandering together with the wind speed deficit gradient has a very high impact on the loads of the downstream turbine, so that a more accurate description in the DWM model with the help of the lidar measurements has high

potential to decrease the uncertainties in load simulations and thus is worth to be investigated.

  Hereafter, in Section 2, a detailed description of the examined wind farm as well as the installed measurement equipment are presented. The filtering and processing of the measured data are explained in Section 3. Section 4 introduces the load simulation software. A specification of the used models as well as the procedure of incorporating the lidar measurements into the model are given in Sections 5 and 6. The document will be completed with the discussion of the results in Section 7 and a

brief summary in Section 8.

## 2 Wind farm and measurement equipment

The analyzed wind farm is located in the southeast of Hamburg, Germany. The terrain is mostly flat and no further wind farms are located in the immediate vicinity. Only in a distance of more than $1\,\mathrm{km}$ the terrain becomes slightly hilly (approx. $40\,\mathrm{m}$ difference in altitude). The wind farm layout is depicted in Figure 1. It includes five closely spaced Nordex turbines (1x N117

3 MW and 4x N117 2.4 MW). All turbines have a hub height of $120\,\mathrm{m}$.



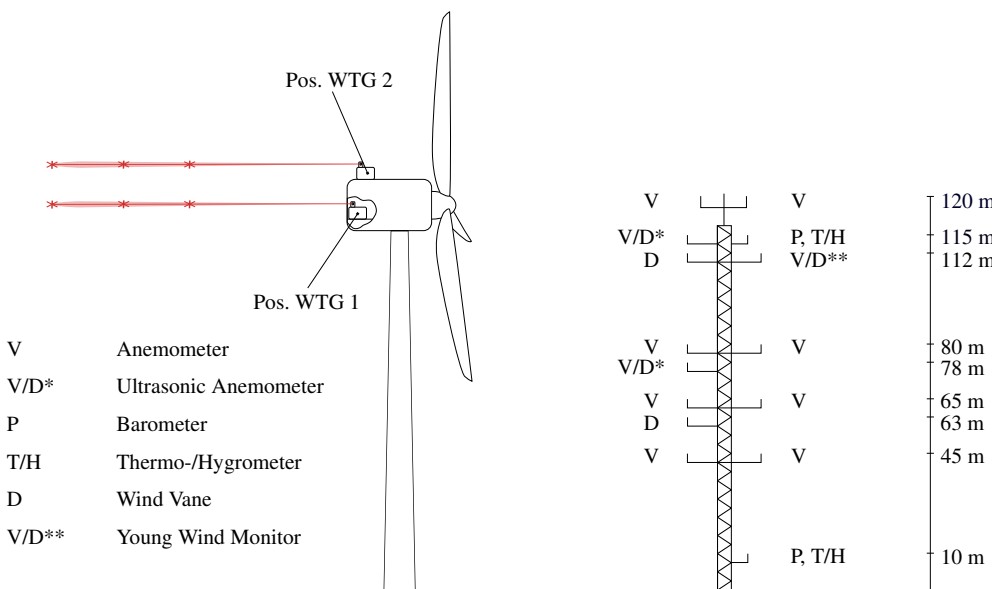

**Figure 2.** Met mast (MM) measurement equipment and lidar positions (Reinwardt et al., 2020).

An IEC compliant $120\,\mathrm{m}$ met mast (IEC 61400-12-1) is placed in main wind direction ahead of the wind farm. It is equipped with 11 anemometers, two of which are ultrasonic devices, three wind vanes, two temperature sensors, two thermohygrometers, and two barometers. The sensors are distributed along the whole met mast and depicted in Figure 2. Furthermore, the turbine nacelles of WTG 1 and WTG 2 are each equipped with a pulsed scanning lidar system (Galion G4000) with a pulse repetition
rate of $15\,\mathrm{kHz}$ and a ray update rate of $\sim 1\,\mathrm{Hz}$ (depending on the atmospheric conditions), so that an average value of approximately 15000 pulses is used per sample. The laser frequency is at $100\,\mathrm{MHz}$. Considering the speed of light, this delivers a pulse length of $1.5\,\mathrm{m}$. Hence, with a range gate length of $30\,\mathrm{m}$, 20 points are used per range gate. Both lidar systems face downwind as depicted in Figure 2. The device on WTG 2 is installed on top of the nacelle, whereas the device on WTG 1 is installed inside the nacelle, measuring through a hole in the rear wall. The unusual location derives from the fact that a recuperator on
top of the nacelle occupies the esssential mounting area. Additionally, nacelle-mounted differential GPS systems help tracking the nacelle's precise position with a centimeter range accuracy, so that yaw movements can be calculated.

At last, three turbines are equipped with load measurements. The tower top and bottom as well as blade flapwise and edgewise bending moments are measured with strain gauges at WTG 2 and WTG 5. WTG 3 is only equipped with strain gauges at the tower. The strain gauges at the tower top are installed $3.4\,\mathrm{m}$ below the nacelle and the strain gauges at the
tower bottom are placed $1.5\,\mathrm{m}$ above the floor panel. The edgewise and flapwise moments are measured in a distance of $1.5\,\mathrm{m}$ from the blade root. Besides the installed measurement equipment, the turbine's Supervisory Control and Data Acquisition (SCADA) system is used to determine the operational conditions of the turbines.



## 3 Data filtering and processing

Measurement results were analyzed from April 2019 to May 2020. The data are filtered and sorted in accordance with the ambient conditions (e.g., ambient wind speed, turbulence intensity and wind direction) determined by the met mast and the operational states of the turbine tracked by the SCADA system, so that all filtering is based on 10-minute statistics from the met mast or the SCADA system. Only measurement results with normal power production are included in the analysis. In the night the turbines working in a reduced mode for noise reduction purposes, so that no data could be gathered during the night. The filtering procedure leads to a high decrease of available data sets, so that e.g. at a turbulence intensity of $6\,\%$ and

**Table 1.** Considered wind direction sectors for wake-free inflow and analyzed wake sectors.

|  | lower limit [°] | upper limit [°] |
| --- | --- | --- |
| Wake-free inflow met mast & WTG 2 | 140 | 260 |
| Wake at WTG 2 generated by WTG 1 | 259 | 335 |
| Wake at WTG 5 generated by WTG 2 | 193 | 237 |
| Wake at WTG 5 generated by WTG 1 | 228 | 268 |

an ambient wind speed of $6\,\mathrm{m/s}$ only around 100 10-min data sets could be collected when WTG 2 is placed in the wake of WTG 1. Considerably more data sets could be collected for wake-free inflow conditions.

The measured lidar data are filtered by the power intensity, which is closely related to the signal-to-noise ratio (SNR) of the measurements. Furthermore, the scan time is observed, so that only results with a sufficient scan time to track the wake meandering are considered. Lidar systems measure the line of sight (LOS) velocity. The wind speed in downstream direction is calculated from the lidar's LOS velocity and the geometric dependency of the position of the laser beam relative to the main flow direction as outlined in Machefaux et al. (2012). Thus, the horizontal wind speed is defined as

$$U(t) = U_{LOS} \cdot \frac{1}{\cos(\theta) \cdot \cos(\phi)} \ . \tag{1}$$

where $\theta$ is the azimuth angle and $\phi$ the elevation angle of the lidar scan head. This approach is suitable for small scan opening angles of the scan head like in the measurement campaign presented here. The lidar system is capable of scanning a two-dimensional wind field in different downstream distances simultaneously. Here, the purpose of the lidar system is to capture the meandering and to estimate the wind speed deficit in the MFR. To ensure that the meandering as well as the wind speed deficit in the Horizontal Meandering Frame of Reference (HMFR) can be covered, a horizontal line is scanned instead of a full two-dimensional wind field. The one-dimensional scan consists of only 11 scan points scanned in a horizontal line from $\theta = -20°$ to $20°$ in $4°$ steps. The duration of the horizontal line scan is around $16\,\mathrm{s}$ depending on the visibility conditions during the scan.



## 4 Load simulation

The loads are simulated with the commercial software alaska/Wind (Zierath et al., 2016), which is based on a flexible multibody system. It is an extension of the classical multibody approach where the system consists of rigid bodies connected by joints and force elements. The system is extended by flexible bodies with small deformations. The rigid body motions are vectorially
superimposed with the deformation of the flexible body. The equations of motion are a set of ordinary differential equations. The model consists of submodels for blades, controller, nacelle, pitch system, gearbox, main shaft, high-speed shaft, generator, hub, yaw drive, and foundation. Blades and tower are reduced by a modal superposition of the first 4 eigenmodes. Both submodels are based on finite-element models consisting of Timoshenko beams.

    The multibody model is connected to an aerodynamic code, which includes the blade element momentum (BEM) theory
(Burton, 2011) and delivers aerodynamic forces and moments at the individual blade sections based on the position and velocity of the blade elements provided by the multibody simulation. The classical BEM therory is extended to include dynamic inflow and dynamic stall effects.

    Furthermore, the multibody model is connected to a controller, which uses the generator speed and the pitch angle from the multibody simulation to calculate the generator torque and the pitch velocity and returns them to the multibody model. The
controller used for the simulations is the actual controller implemented in the turbines of the analyzed wind farm. Hence, a reliable comparison with the measured loads can be achieved.

    The inflow wind conditions can be divided into deterministic and stochastic contributions. Deterministic contributions, like the mean wind speed and the shear effects, are imposed on the turbulent wind field. The stochastic contributions are simulated based on a Kaimal spectrum and a coherence function (e.g., Veers, 1988). The DWM model is a standalone in-house tool
written in Python and is uncoupled from the alaska/Wind software. The script generates binary wind files with wake effects, which can be included in alaska/Wind similar to conventional stochastic wind fields.

    The following analysis covers simulated power, blade root flapwise and edgewise bending moments as well as tower bottom bending moments. Auxiliary sensors are added to the turbine model in alaska/Wind to compare the measured loads at the precise position of the strain gauges, the locations of which are given in Section 2.

## 5 Dynamic wake meandering model

The measured loads under wake conditions are compared to the simulated loads, which incorporate the DWM model to simulate the inflow at the wake-affected turbine. As mentioned before, the DWM model is based on the assumption that the wake behaves like a passive tracer in the turbulent wind field. Consequently, the movement of the passive structure, i.e., the wake deficit, is driven by large turbulence scales (Larsen et al., 2007, 2008b). The main components of the model are summarized in
Figure 3.

    One part of the model is the quasi-steady wake deficit, or rather the wind speed deficit in the MFR, which consists of a definition of the initial deficit emitted by the wake generating turbine and the degradation of the deficit downstream (Larsen et al., 2008a). The expansion in downstream direction is calculated with the thin shear layer approximation of the Navier-Stokes





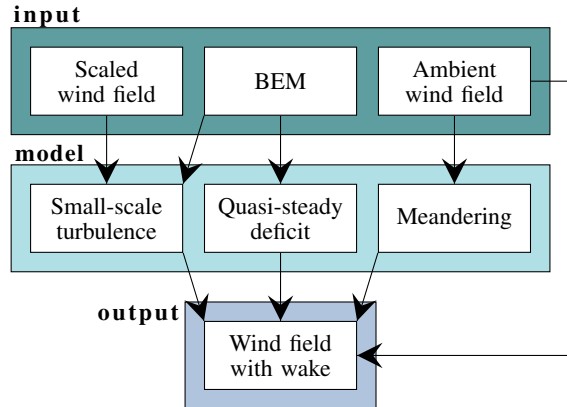

**Figure 3.** Components of the DWM model (adapted from Reinwardt et al., 2018).

equations in their axisymmetric form and thus is strongly related to the work of Ainslie (1988). The method in the DWM model
is outlined in Larsen et al. (2007). Using a finite-differences method combined with an eddy viscosity ($\nu_T$) closure approach,
the thin shear layer equations are solved directly starting at the rotor plane. The emitted initial deficit serves as a boundary
condition when solving the equations. It is based on the axial induction factor derived from the BEM theory. Three calculation
methods of the quasi-steady wake deficit, which differ only in the description of the initial deficit and the eddy viscosity, will
be compared in the course of this study:

– "DWM-Egmond" based on the definitions in Madsen et al. (2010) and Larsen et al. (2013),

  – "DWM-Keck" adopted from Keck (2013) and

  – "DWM-Keck-c", a recalibrated version of the "DWM-Keck" model based on lidar measurements from the wind farm
    underlying here (Reinwardt et al., 2020).

A detailed description of the individual models can be found in Reinwardt et al. (2020).

Another aspect of the model is the description of the wake meandering. In this work it is calculated from the large turbulence
scales of the ambient turbulent wind field, which is generated by a Kaimal spectrum and a coherence function (e.g., Veers,
1988) and subsequently ideally low-pass filtered. Afterwards, the vertical and horizontal movements are determined based on
the filtered wind field. The cut-off frequency of the low-pass filter is specified by the ambient wind speed and the rotor diameter
(Larsen et al., 2013).

The third part of the DWM model is the definition of the small-scale turbulence generated by the wake shear itself as well as
by blade tip and root vortices. This small-scale turbulence is calculated with a scaled homogeneous turbulent wind field, which
is also generated by a Kaimal spectrum. The scaling is implemented in accordance with IEC 61400-1 Ed.4. A more detailed
description of the implementation of the complete model can also be found in Reinwardt et al. (2020).





## 6 Lidar assisted load simulation

In the previous section, a recalibrated version of the DWM model has been introduced. The lidar systems have been used to recalibrate the DWM model to decrease the uncertainties of load simulations in wake conditions. In a next step, the lidar measurements will be successively incorporated into the wake simulation. A schematic illustration of the process is illustrated in Figure 4.

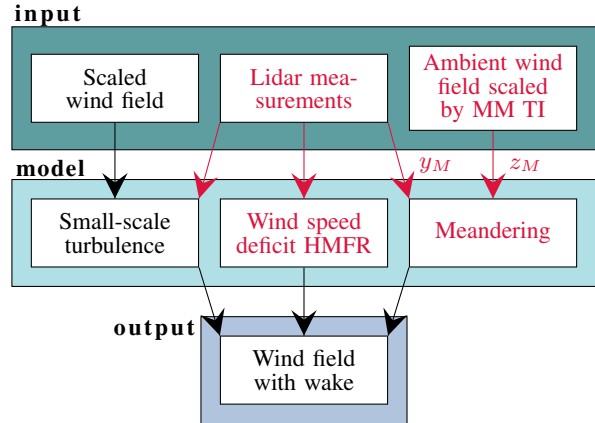

**Figure 4.** Incorporation of lidar measurements into the DWM model; $y_M$ is the horizontal and $z_M$ the vertical meandering component.

First, the lidar-measured mean wind speed deficit is used to replace the quasi-steady deficit in the DWM model definition
(see also Figure 3). Since only a horizontal line is scanned, no vertical meandering can be captured (see Section 3). To clarify that only the horizontal meandering can be measured and that the transformed wind speed deficit in the meandering frame of reference still includes the vertical meandering, the phrasing Horizontal Meandering Frame of Reference (HMFR) is introduced in Figure 4. In a second step, the measured horizontal meandering is included in the DWM model and the vertical meandering has been neglected. The vertical meandeing has only a marginal influence in the shape of the deficit in the MFR as explained
in Reinwardt et al. (2020).

The lidar system measures in the induction zone of the downstream turbine, where the wind speed is decreased due to the upstream effect of the subsequent turbine. However, its influence must be excluded from the measurement results to use the measured wind speed deficit in the wake model. The simple induction model defined in Troldborg and Meyer Forsting (2017) is applied to account for this effect. The two-dimensional model defines the wind speed in the induction zone as follows:

$$U = U_0 \left[ 1 - a_0 \left( 1 - \frac{\tilde{x}_u}{\sqrt{1 + \tilde{x}_u^2}} \right) \left( \frac{2}{\exp(+\beta\epsilon) + \exp(-\beta\epsilon)} \right)^{\alpha_i} \right], \tag{2}$$

where $\tilde{x}_u$ is the positive upwind distance normalized by the rotor radius, $a_0$ is the induction factor at the rotor center area defined as $a_0 = 0.5(1 - \sqrt{1 - \gamma c_t})$, $\epsilon = \tilde{r}/\sqrt{\lambda(\eta + \tilde{x}_u^2)}$, $\tilde{r}$ is the radial distance from the hub normalized by the rotor radius, $c_t$ is the thrust coefficient, $\gamma = 1.1$, $\beta = \sqrt{2}$, $\alpha_i = 8/9$, $\lambda = 0.587$ and $\eta = 1.32$. The model has already been used to correct lidar measurements in the induction zone by Dimitrov et al. (2019) and Conti et al. (2020).





The time series of the meandering and the horizontal displacement of the wake are determined with the help of a Gaussian fit in accordance with Trujillo et al. (2011), who assume that the probability of the wake position in vertical and horizontal direction is completely uncorrelated. The Gaussian function has been fitted to the wind speed deficit, so that the center of the wake could be determined in accordance with the fitting parameters. Since the vertical meandering is neglected in the present case, the measurement results are fitted to a one-dimensional Gaussian curve:

$$f_{1D} = \frac{A_{1D}}{\sqrt{2\pi}\sigma_y} \exp\left(-\frac{1}{2}\frac{(y_i - \mu_y)^2}{\sigma_y^2}\right) , \qquad (3)$$

where $A_{1D}$ is a scaling parameter and $\sigma_y$ is the standard deviations of the horizontal displacement $\mu_y$. Determining the measured mean wind speed deficit in the HMFR can be summarized as follows:

1. Correction of the measured wind speed by the induction zone model

2. Fitting of a Gaussian curve to the wind speed distribution along the horizontal direction determined by a measured horizontal line scan and determination of the horizontal displacement of the wake

3. Transfer of the measured wind speed deficit to the HMFR by shifting the scan points according to the determined displacement

4. Interpolation of the scanned wind speed deficit in the HMFR to a regular grid

5. Repetition of steps 1 to 4 until a certain number of scans is reached (e.g., approx. 37 for a 10-min time series)

6. Calculation of the mean wind speed deficit in the HMFR from all scans

7. Fitting of the measured mean wind speed deficit to the Bastankhah wake model described in Bastankhah and Porté-Agel (2014)

It should be pointed out that always the closest available measured range gate, which is still outside the rotor area of the downstream turbine, is used to determine the inflow wind speed deficit. Furthermore, the fourth step of interpolating the wind speed deficit to a regular grid is mandatory due to the fact that the horizontal displacement differs at each instant in time and, thereupon, the measurement points are transmitted to a different location in the HMFR, so that the last step of calculating a mean wind speed deficit over all scans is only possible after interpolating all scans to the same regular grid. A more detailed explanation of calculating the wind speed deficit in the HMFR can be found in Reinwardt et al. (2020).

An example of the measured and simulated time series of the meandering as well as the power spectrum is shown in Figure 5. It depicts the measured time series of the meandering as well as the one simulated with the Keck-c model and a random turbulence seed. To incorporate the time series of the meandering in the wake and load simulations, the time series has been cubically interpolated, so that a smooth meandering could be included in the wake model and the turbine loads are not increased by an immediate change of the position of the wind speed deficit. The interpolated time series of the meandering is denoted as DWM-meas. The comparison of simulations and measurements shows that the amplitude of the measured time



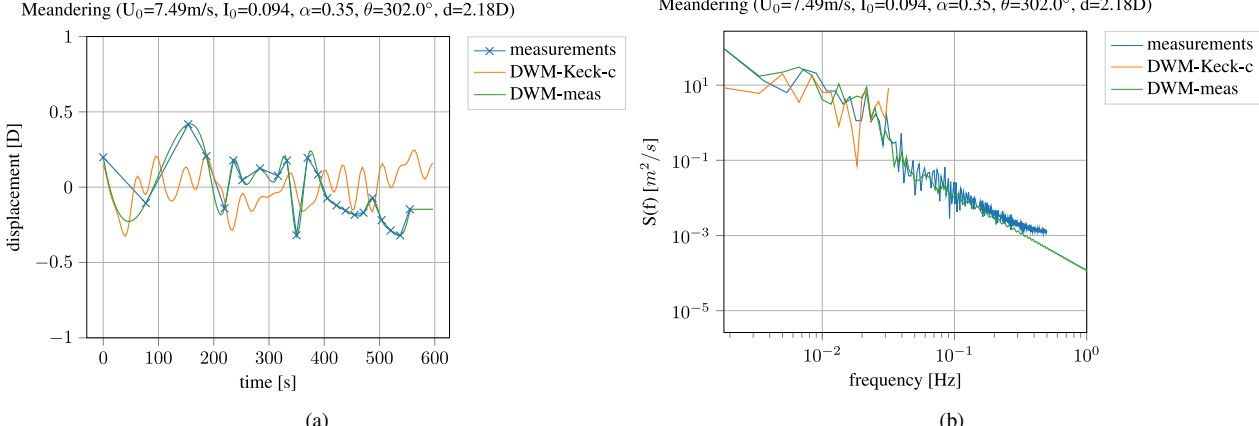

**Figure 5.** Time series (a) and power spectrum (b) of the meandering; measured and simulated with the calibrated DWM-Keck-c model as well as the interpolated time series (DWM-meas).

series is more pronounced. A reason could be that the meandering is modeled based on the ambient wind speed although the wind speed in the wake is reduced. Applying a reduced mean wake wind speed in the meandering calculation procedure would lead to a higher deflection of the wake. It should also be pointed out that the measurement frequency is very low due to the data filtering in the beginning, so that it might be the case that some parts of the meandering could not be captured by the measurements.

An example of a measured wind speed deficit over the radial distance from the hub center in the HMFR in comparison to the simulated one with the recalibrated DWM model is illustrated in Figure 6. The ambient conditions (ambient wind speed $U_0$, ambient turbulence intensity $I_0$, wind shear $\alpha$ and wind direction $\theta$) are defined in the title of the figure. The edges of the measured deficit are coarser than the area close to the center of the deficit. The explanation for this observation is as follows. The distribution generated by the meandering process provides many scan points around the center of the wind speed deficit

and only a few at the tails, so that the influence of turbulence at the tails is much higher, which is why the measured wind speed deficit shows a coarse distribution at the boundaries of the deficit. Using this coarse curve leads to increased loads in the simulation, which are not feasible, hence the mean wind speed deficit in the HMFR in the DWM model definition should be replaced by the lidar measurements. Furthermore, the lidar system only measures an opening angle of $-20°$ to $20°$. Hence, particularly for short distances the deficit is not captured exhaustively. Even the ambient wind speed is not reached at the

edges of the curve, thus it is necessary to extrapolate the wind speed to smoothly meet the ambient wind speed. As a result of these issues, the measured deficit has been fitted to a simple Gaussian shaped wake model (Bastankhah model) outlined in Bastankhah and Porté-Agel (2014). According to the model, the wind speed deficit can be defined as:

$$\frac{\Delta U}{U_0} = \left(1 - \sqrt{1 - \frac{c_t}{8\left(2k^*\tilde{x} + 0.2\sqrt{\beta}\right)^2}}\right) \exp\left(-\frac{c_t}{2\left(2k^*\tilde{x} + 0.2\sqrt{\beta}\right)^2} 4\left((\tilde{z} - \tilde{z_h})^2 + \tilde{y}^2\right)\right) \tag{4}$$





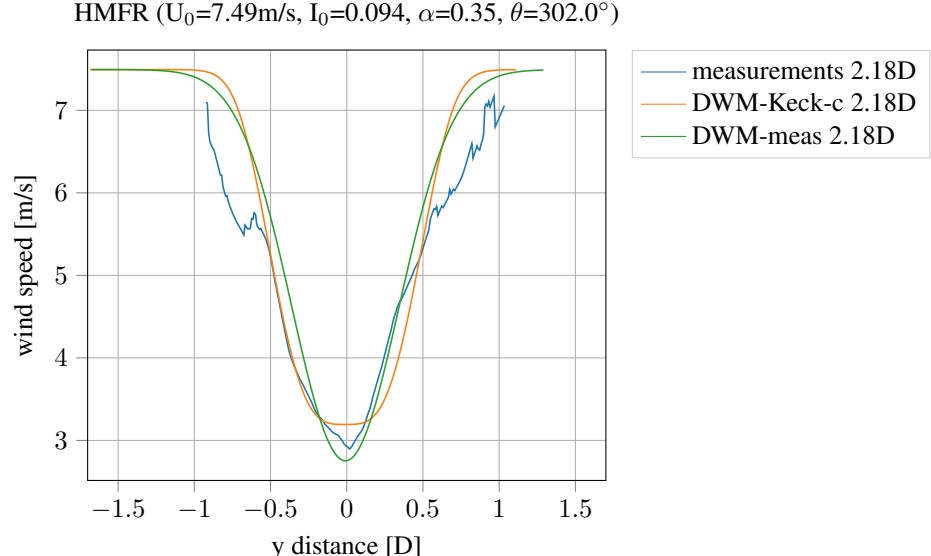

**Figure 6.** Wind speed deficit in the HMFR; measured and simulated with the calibrated DWM-Keck-c model as well as fitted to a Gaussian shaped wake model (DWM-meas).

with $k^*$ being the wake growth rate, $\tilde{x}$ the downstream distance normalized by the rotor radius, $\tilde{z}_h$ the normalized hub height, $\tilde{y}$ and $\tilde{z}$ the normalized horizontal and vertical distance and

$$\beta = \frac{1}{2}\frac{1+\sqrt{1-c_t}}{\sqrt{1-c_t}}. \tag{5}$$

The wake growth rate $k^*$ has been adjusted to fit the model to the measured deficit in the HMFR. The fitted model is labeled "DWM-meas" in Figure 6.

## 7 Results

### 7.1 Comparison of measured and simulated loads and power under wake-free inflow

In order to validate the aerodynamic load simulations the following section contains a comparison of measured and simulated loads under wake-free inflow conditions. The section shows results from WTG 2 under normal operating conditions. The met mast as well as WTG 2 are exposed to wake-free inflow conditions, thus the met mast is suitable to determine all ambient conditions. The mean value of the measured and simulated normalized power curve is depicted in Figure 7(a) for a turbulence intensity of $12\,\%$.

The power curve is normalized by the measured power in the smallest wind speed bin. The error bars in the curves illustrate the standard deviation in each wind speed bin. All data sets are divided into wind speed bins with a width of $1\,\mathrm{m/s}$. The mean values of wind speed, turbulence intensity, wind shear, and air density of each wind speed bin determine the input parameters

**Figure 7.** Measured and simulated power (a), flapwise blade root moment (b), edgewise blade root moment (c) and tower bottom fore-aft moment (d) at WTG 2 at an ambient turbulence intensity of 12 % and wake-free inflow.

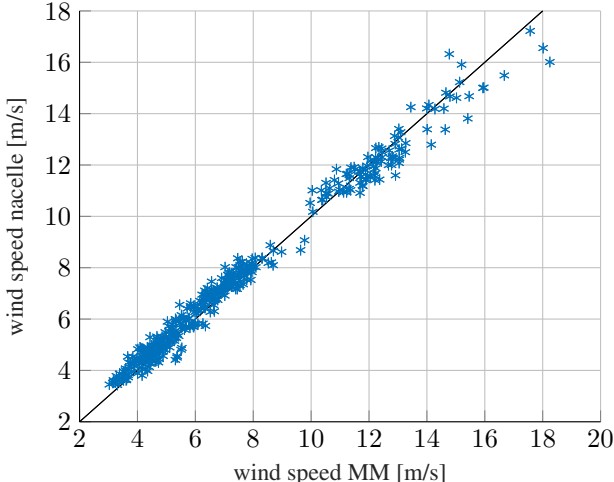

**Figure 8.** Measured nacelle anemometer wind speed at WTG 2 over met mast wind speed.

for the load simulations. Each simulation is conducted six times with different seeds, so that the simulation results are likewise

shown as a mean values with standard deviations. In summary, the simulated power agrees very well with the measured power, solely close to the rated wind speed some discrepancies between measurements and simulations occur. In this area only a few measurement points can be extracted due to the chosen filtering criteria. As a result, the measurements show an extraordinarily high standard deviation. A comparative study of the measured wind speed of the nacelle anemometer and the met mast has indicated some discrepancies in this range (see Figure 8). Thus it is very likely that the deviation arises due to a momentarily

different inflow wind speed at the turbine than the one measured at the met mast and used in the simulations. The measured wind speed at the nacelle anemometers is corrected by a nacelle transfer function, so that the current inflow wind speed at the turbine can be estimated.

The results of the measured and simulated flapwise blade root bending moment are illustrated in Figure 7(b). It displays the normalized 1 Hz damage equivalent load (DEL). The development of the measured flapwise fatigue load as a function of the

wind speed can be reproduced very well by the simulations. Only some slight discrepancies occur between $6\,\mathrm{m/s}$ and $9\,\mathrm{m/s}$, where the simulation overestimates the loads slightly. The measured and simulated DEL of the edgewise blade root bending moment is depicted in Figure 7(c). The simulations of the edgewise moment show a local maximum around the rated wind speed. This observation could not be verified by the measurements. The measured and simulated power deviate in this wind speed range, which can be explained by differences between the nacelle anemometer and the met mast anemometer in the

estimated wind speeds. It is most likely that the load discrepancies in this range derive from the same issue. The differences in the edgewise moment and the power around rated wind speed as well as the illustration of the measured nacelle wind speed and the met mast prove the fact that the turbine experiences a local momentarily different inflow wind speed and explains the discrepancies. However, since the differences between measurements and simulations in the edgewise moment are still below $5\,\%$, the overall agreement is reasonable for this load component. The edgewise moment is mainly driven by the rotational speed





of the rotor and the gravity. The dependency of the edgewise moment on the wind speed is less pronounced in comparison to
the flapwise moment. The simulated DELs of the tower bottom bending moment are depicted in Figure 7(d). The measured
tower bottom bending moment can be predicted very well by the simulation, although there are similar discrepancies around
the rated wind speed. Nevertheless, the accuracy of the used load simulation software in combination with the turbine model
for load simulations are presumed to be appropriate for a further analysis of the wake sectors, since for the wake analysis only
results below the rated wind speed are analyzed.

### 7.2    Comparison of measured and simulated loads and power under wake conditions

### 7.2.1    Analysis at an ambient wind speed of $6 \, \mathrm{m/s}$ and a turbulence intensity of $6 \, \%$

The following section summarizes the measured and simulated fatigue loads under normal operation conditions for the wake
sector. The ambient conditions for the simulations are determined by the met mast, so that only results with wake-free inflow at
the met mast are included in the evaluation. The results of the measured and simulated normalized power deficit, where WTG 2
experiences the wake of WTG 1, are shown in Figure 9(a).

The results are normalized by the measured power at wake-free inflow on the left side of the power deficit curve. The mea-
surements were gathered during an ambient wind speed of $6 \, \mathrm{m/s}$ with and an ambient turbulence intensity of $6 \, \%$. The mean
values and their corresponding standard deviations are illustrated for each wind direction bin, along with the number of consid-
ered measurements and simulations. Close to full wake, only a few measurement points could be collected, whereas towards
the edges of the deficit more points could be gathered. The reason is that the deficit towards full wake is very pronounced
and thus the inflow wind speed at the wake-affected turbine is often below the cut-in wind speed. Three different versions of
the DWM model are used in the simulations as introduced in Section 5. All simulated power deficits agree very well with the
measured deficit. There is a slight overestimation of the power deficit calculated by the DWM-Egmond model. Its predicted
deficit is so pronounced that in full wake conditions the turbine often does not operate in the simulations. There is a slight
decrease of the power above 320°, which marks the beginning of the wake of WTG 4.

The DELs of the flapwise blade root bending moment under wake conditions are illustrated in Figure 9(b). The flapwise
fatigue loads agree very well with the measurements when using the DWM-Keck-c and DWM-Keck model, so that even the
two maxima at partial wake conditions are in close agreement. The DWM-Egmond model overpredicts the loads especially at
partial wake conditions. At partial wake, the wind speed deficit only affects a section of the rotor, so that in combination with
the meandering and the horizontal shear of the wind speed, the blade experiences a high alternating load with each rotation.

The two maxima are differently pronounced, which derives from the aerodynamic force and the rotor tilt. Because of the
tilt, the blade faces slightly away from the wind direction during the upward movement and the aerodynamic force is reduced,
whereas during the downward movement, the blade faces slightly more towards the wind direction, which results in an increase
of the aerodynamic force. At wake conditions, the increase is stronger when the wind speed deficit coincides with the upward
movement of the rotor, so that a higher alternating load at the blade occurs and the maximum is more pronounced in comparison
to the case where the wind speed coincides with the downwind movement of the rotor.

**Figure 9.** Measured and simulated power deficit (a), flapwise blade root moment (b), edgewise blade root moment (c) and tower bottom fore-aft moment (d) at WTG 2 at an ambient wind speed of 6 m/s and an ambient turbulence intensity of 6 %. WTG 2 is exposed to the wake of WTG 1.



The results of the edgewise blade root bending moment are depicted in Figure 9(c). All models agree similarly well with the measurements. The edgewise moment depends significantly on the blade weight force, while the wake only has a marginal

impact on the loads, so that the highest increase of the edgewise moment in comparison to wake-free inflow is merely around 5 %. Towards full wake, several outliers in the DWM-Keck and DWM-Egmond model can be recognized. These are related to the simulations, where the turbine does not operate as a result of the low wake wind speed predicted by the models. The rotation of the rotor largely influences the alternating load at the edgewise moment, hence the fatigue load is drastically reduced when the turbine turns off. The simulations as well as the measurements show an increase of the load in comparison to the wake-free

inflow at around 280° and even a decrease of the load at around 310°. The influence of the wind speed is not only related to the rotational speed of the rotor. There is an additional influence due to the tilt of the rotor. The load is defined in the rotating frame of reference, so that the weight force switches its sign with each rotation, whereas the influence of the aerodynamic force on the edgewise moment does not change the sign. Thus, at one side of the rotor the forces level each other out, while on the other side of the rotor they accumulate. If the deficit is on the side where the forces level each other out, the alternating load increases

in comparison to a situation without wake, whereas when the wind speed deficit is on the side where both aerodynamic and weight force are facing in the same direction, the alternating load is decreased.

The tower bottom bending moment is illustrated in Figure 9(d). The two maxima at partial wake conditions that derive from the higher alternating load observed in Figure 9(b) are clearly visible for the tower bottom bending moment, too. At full wake conditions the load is only slightly increased in comparison to wake-free inflow. Though, similar to the flapwise moment, the

tower bottom bending moment is almost doubled at partial wake conditions. The results of all three models agree well with the measurements. Only the DWM-Egmond model overestimates the loads as it could already be seen in the blade flapwise and edgewise moments.

### 7.2.2 Comparison of different turbulence intensities

A similar analysis as the one presented in Figure 9 is carried out for different turbulence intensity bins and summarized

in Figure 10. A comparison of the results of the flapwise bending moment over all turbulence intensity bins is shown in Figure 10(a). It illustrates the bias of the accumulated DEL over all wind directions. A negative value implies a lower value of the simulated accumulated DEL than the measured DEL. The accumulated DEL over all wind directions is calculated with respect to the Wöhler coefficient. A Wöhler coefficient of 10 is used for the blades and 4 for the tower loads as specified in the titles. Of all models, the recalibrated DWM-Keck-c model coincides best with the measurements over all turbulence

intensity bins. At small turbulence intensities, the DWM-Keck-c model underestimates the accumulated DEL slightly. The DWM-Egmond model overestimates the accumulated DEL drastically, especially at high turbulence intensities. The root-mean-square error (RMSE) between the simulation and the measurement over the wind directions is given in Figure 10(b). The RMSE of the DWM-Keck and the recalibrated DWM-Keck-c model are very similar, while the DWM-Egmond delivers the highest RMSE. The reason for illustrating the deviation between measurements and simulations as well as the RMSE is that

the deviation of the accumulated DEL expresses how accurate the models perform in a site-specific load calculation procedure. Additionally, it allows a comparison with the Frandsen model (Frandsen, 2007), whereas the RMSE represents the overall





**Figure 10.** Bias between the measured and simulated fatigue loads and the RMSE for the flapwise blade root bending moment (a) and (b), the edgewise blade root bending moment (c) and (d) as well as the tower bottom bending moment (e) and (f) at an ambient wind speed of $6\,\mathrm{m/s}$. WTG 2 is exposed to the wake of WTG 1.





capability of predicting the distribution of the DEL over the wind direction. Note that the Frandsen model overestimates the DELs significantly throughout all turbulence intensities.

Figure 10(c) depicts the accumulated DELs and the RMSE of the edgewise blade root bending moment. The smallest devia-
tion between the accumulated DELs is achieved with the DWM-Egmond model, but the difference between the models is very small, so that even the highest deviation with the DWM-Keck-c model is only about $1.4\,\%$. The RMSE of the DWM-Keck-c model is the lowest (see Figure 10(d)). However, it should be pointed out that also the RMSE is very low in all cases. The re-sults over different turbulence intensity bins of the tower bottom fore-aft bending moment are shown in Figure 10(e-f). Similar to the flapwise moment, the accumulated DEL over all wind directions calculated by the DWM-Keck-c model agrees very well
with the measurements. Again, only a slight underestimation occurs at small turbulence intensities. The DWM-Egmond as well as the Frandsen model overestimate the accumulated DEL substantially. The RMSE of the recalibrated and the original model are low and have similar magnitudes. The DWM-Egmond model delivers the highest RMSE over all turbulence intensity bins.

### 7.2.3 Comparison of different downstream distances

This section compares results for different downstream distances at an ambient wind speed of $8\,\mathrm{m/s}$ and an ambient turbulence
intensity of $10\,\%$. The following three wake situations have been analyzed:

- WTG 2 in the wake of WTG 1 $\rightarrow$ turbine distance $= 2.51D$

- WTG 5 in the wake of WTG 2 $\rightarrow$ turbine distance $= 3.61D$

- WTG 5 in the wake of WTG 1 $\rightarrow$ turbine distance $= 4.71D$

The results of the power deficit over the wind directions for these three different distances are shown in Figure 11. The plots

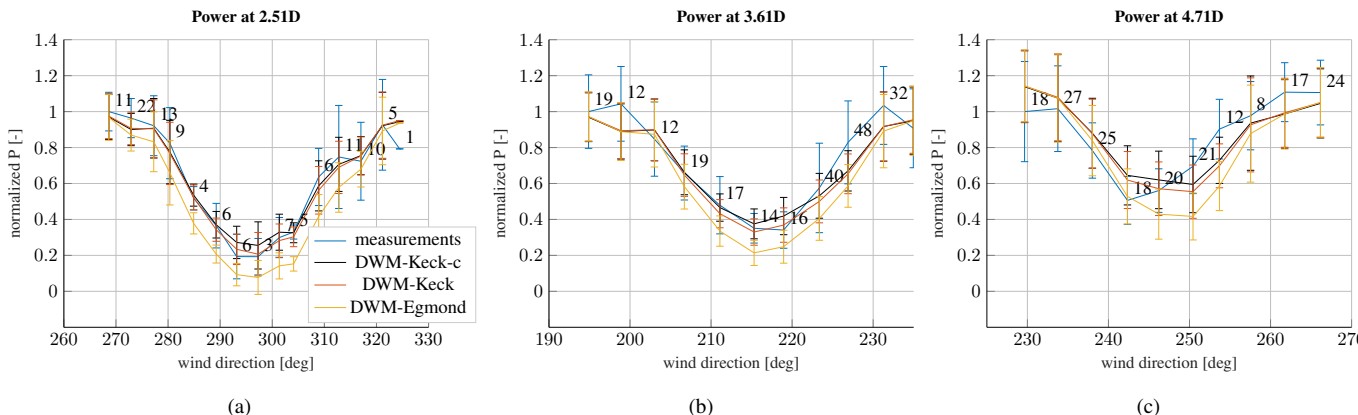

**Figure 11.** Measured and simulated power at an ambient wind speed of $8\,\mathrm{m/s}$ and an ambient turbulence intensity of $10\,\%$ when WTG 2 is exposed to the wake of WTG 1 (a), WTG 5 is exposed to the wake of WTG 2 (b), and WTG 5 is exposed to the wake of WTG 1 (c).

display the mean value in each wind direction bin accompanied by the corresponding standard deviation as an error bar. The



same DWM model versions as previously discussed are compared. The notation of the models and the basic structure of the plots follow the figures of Section 7.2.1. The closest turbine distance of $2.51D$ shows the most pronounced deficit and vice versa. The predicted results of all models agree very well with the measurements. The DWM-Egmond model overestimates the deficit, especially at the largest distance of $4.71D$.

The results of the flapwise and edgewise blade root moments as well as the tower bottom bending moments are shown in Figure 12. The results of the recalibrated and original DWM-Keck models agree very well with the measurements over all distances. The DWM-Egmond model on the other hand overestimates the loads mostly, particularly at the highest distance of $4.71D$. The reason is that the degradation of the wake over the downstream distance is underestimated by this model. The eddy-viscosity definition in the DWM-Keck-c model has been recalibrated by lidar measurements from the site. As a result, a

higher and more suitable degradation of the wake could be achieved.

The bias of accumulated DELs over all wind directions as well as the RMSE are depicted in Figure 13. The recalibrated DWM-Keck model delivers the lowest deviation and RMSE over all distances for the flapwise moment and the tower bottom bending moment, whereas the edgewise blade root bending moment is not improved by the recalibration. However, as mentioned in the previous section, the difference between the results of the single variations of the DWM model is very low for

this load component, so that all models agree very well with the measurements of the edgewise moment with the exception of the Frandsen wake model. The reason for this is probably that no wind speed deficit is considered in Frandsen's model, so that the alternating load at the flapwise moment is higher due to the higher wind speed and the rotational speed of the rotor. The DWM-Egmond model overestimates the loads over all downstream distances. Towards greater downstream distances, the improvement due to the recalibration increases. Lastly, the Frandsen model overestimates the loads over all distances, in

particular for close spacings.

### 7.2.4    Comparison with lidar assisted load simulations

In the following, the recalibrated DWM model is compared to a constrained simulation with lidar measurements of the meandering and the wind speed deficit. The method to incorporate the wind speed deficit in the HMFR as well as the meandering itself is explained in Section 6. Figure 14(a) shows the measured power deficit at WTG 2 when the turbine is exposed to the

wake of WTG 1 at an ambient wind speed of $8\,\mathrm{m/s}$ and an ambient turbulence intensity of $10\,\%$. The blue solid curve with error bars is the measured mean power deficit with all measurement results that comply with the requirements for ambient conditions and the filtering criteria. Lidar measurements were not available for all collected data sets. The blue circles illustrate the 10-min time series where lidar measurements are available. The stars denote the simulated 10-min series using the recalibrated version of the DWM model. The crosses represent the results when incorporating only the measured wind speed deficit in the HMFR

fitted to the Gaussian shaped wind speed deficit model, whereas the squares consider both the measured meandering and the wind speed deficit in the HMFR. The RMSE between measurements (blue circles) and simulations are given in the legend. The recalibrated DWM model and the constrained simulations, which only uses the measured wind speed deficit shape, agree similarly well with the measurements, but the results based on the incorporation of the measured meandering fit considerably better to the measurements, especially towards the left part of the curve. It has been observed that the meandering is more



**Figure 12.** Measured and simulated flapwise blade root bending moment (a)-(c), edgewise blade root bending moment (d)-(f), tower bottom bending moment (g)-(i). WTG 2 is exposed to the wake of WTG 1 in (a),(d), and (g), WTG 5 is exposed to the wake of WTG 2 in (b),(e), and (h), and WTG 5 is exposed to the wake of WTG 1 in (c),(f), and (i). The ambient wind speed is $8\,\text{m/s}$ and the ambient turbulence intensity is $10\,\%$.



**Figure 13.** Bias between the measured and simulated fatigue loads and the RMSE for the flapwise blade root bending moment (a) and (b), the edgewise blade root bending moment (c) and (d), as well as the tower bottom bending moment (e) and (f) at an ambient wind speed of $8\,\text{m/s}$ and an ambient turbulence intensity of $10\,\%$.



WIND
ENERGY
SCIENCE
DISCUSSIONS
eawe
european academy of wind energy
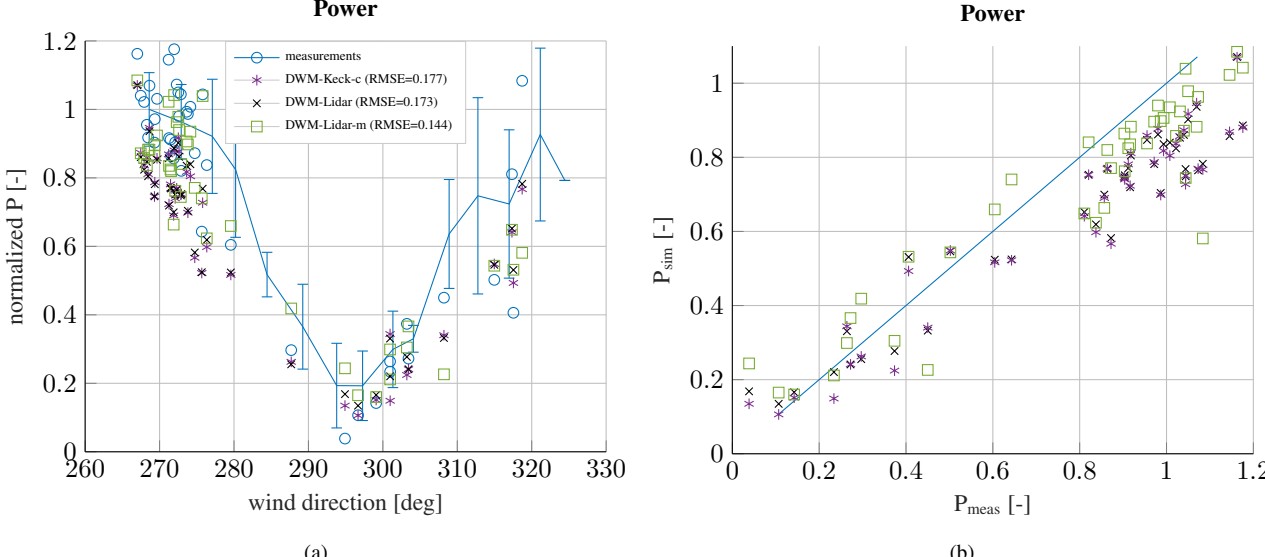

**Figure 14.** Measured and simulated power over the wind direction (a) and simulated power over measured power (b) at an ambient wind speed of $8\,\mathrm{m/s}$ and an ambient turbulence intensity of $10\,\%$, when WTG 2 is exposed to the wake of WTG 1.

pronounced in the measurements than in the DWM model simulations as it could already be seen in Figure 5. Thus, especially at the edges of the wake, when the downstream turbine is almost out of the wake, the amplitude of meandering becomes more important. If the meandering is more pronounced in this region, the wake-affected turbine experiences wake-free inflow conditions more often. Furthermore, if there is a slight misalignment of the wake generating turbine, it is indirectly captured in the determination of the meandering.

The simulated power over the measured power is illustrated in Figure 14(b). The plotted straight blue line has a slope of one and serves as a reference. The underestimation of the power deficit in the simulations is clearly visible in the upper part of the figure, just like the improvement when considering the measured meandering.

The results of the flapwise blade root bending moment are given in Figure 15. A clear overestimation of the loads can be seen in the flapwise moment, so that there is a higher influence of the wake in the simulations. The incorporation of the wind speed deficit leads to a slightly better agreement between measurements and simulations than only using the recalibrated DWM model. Including the time series of the meandering leads to even better coincidences between measurements and simulations. However, the simulations overestimate the loads towards the edges of the curve. A similar behavior can be seen for the edgewise moment as well as the tower bottom bending moment (see Figures 16 and 17), although the difference between simulations and measurements are smaller for these load components. Another explanation for the differences and uncertainties can be found in the downstream distance, which is used in the simulations. For the comparison measurements at the closest available lidar range gate that is still outside the rotor area of the downstream turbine is used, thus it happens that the downstream distance used in the simulations is slightly to low. To achieve a suitable comparison with the DWM-Keck-c model the measurement



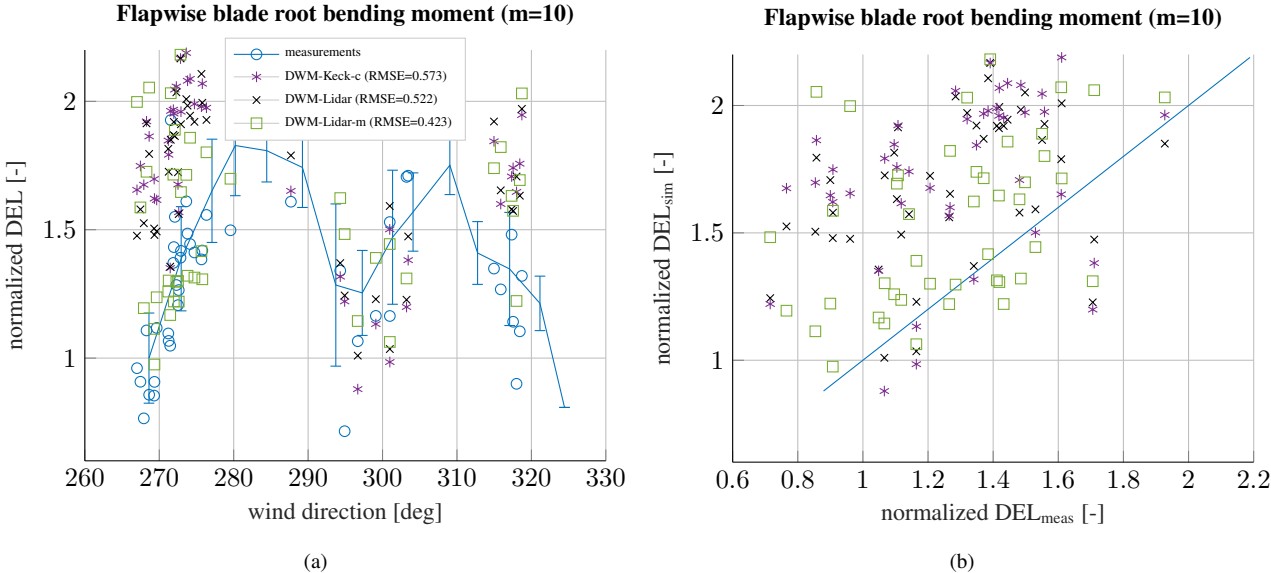

(a)

(b)

**Figure 15.** Measured and simulated flapwise blade root bending moment over the wind direction (a) and simulated loads over measured loads (b) at an ambient wind speed of 8 m/s and an ambient turbulence intensity of 10 %, when WTG 2 is exposed to the wake of WTG 1.

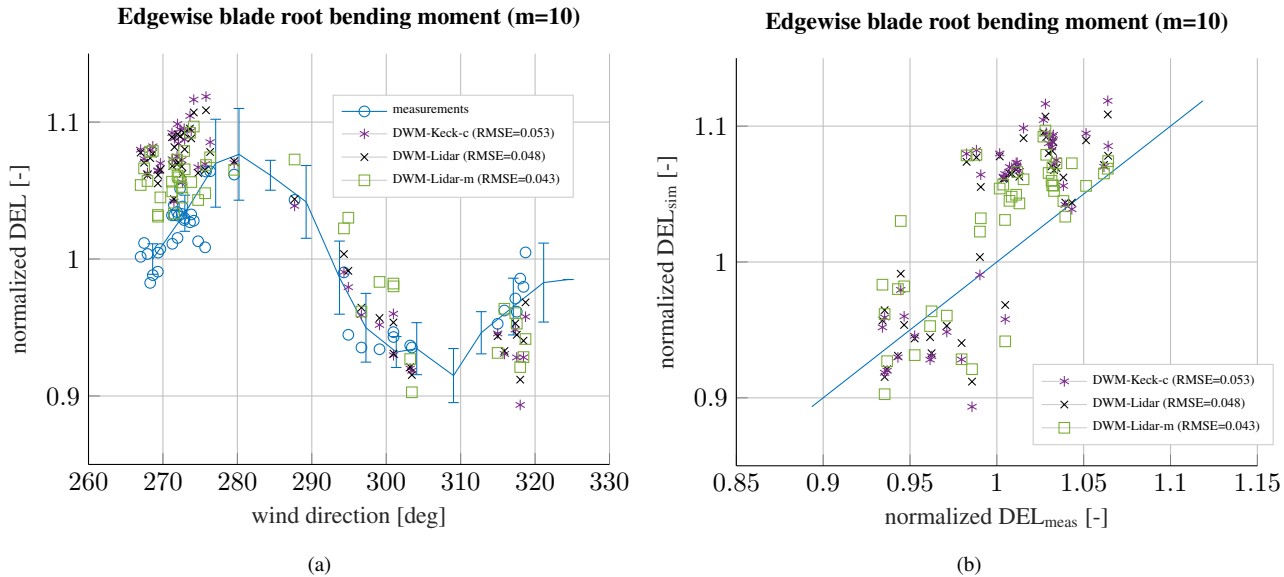

(a)

(b)

**Figure 16.** Measured and simulated edgewise blade root bending moment over the wind direction (a) and simulated loads over measured loads (b) at an ambient wind speed of 8 m/s and an ambient turbulence intensity of 10 %, when WTG 2 is exposed to the wake of WTG 1.



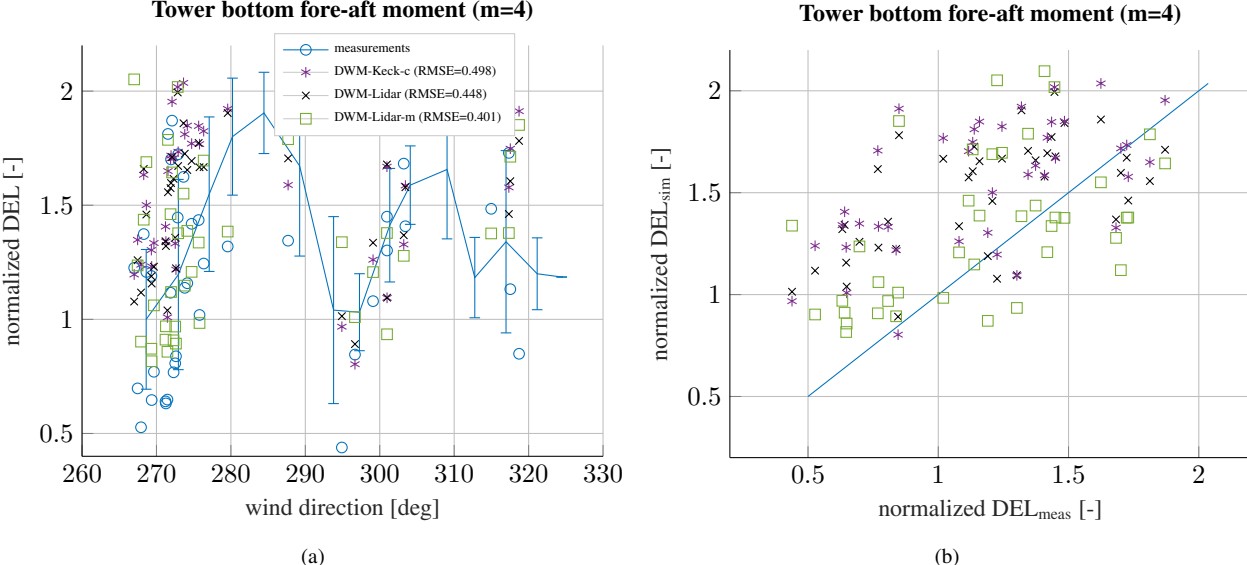

**Figure 17.** Measured and simulated tower bottom fore-aft bending moment over the wind direction (a) and simulated loads over measured loads (b) at an ambient wind speed of $8\,\text{m/s}$ and an ambient turbulence intensity of $10\,\%$, when WTG 2 is exposed to the wake of WTG 1.

distance has been used in the model as well. However, the influence should be small, due to the small gradient of the wind speed in downstream direction. Furthermore, it should be pointed out that the vertical meandering is neglected, when the measured

meandering is used, due to the fact that no vertical meandering is captured by the lidar systems. The vertical movement is less pronounced than the horizontal meandering and has only a small influence on the shape of the wind speed deficit in the fixed frame of reference and the loads. Hence, this simplification barely affects the overall results.

## 8    Conclusions

The outlined analysis validates the DWM model based on power and load measurements at an onshore wind farm with small

turbine distances. Special focus is put on a calibrated version of the DWM model (Reinwardt et al., 2020). The model has been calibrated based on nacelle-mounted lidar measurements. Additionally, a comparison with the commonly used Frandsen model is performed. The newly calibrated model fits very well to measurement results, whereas the Frandsen model delivers very conservative results for short turbine distances. Furthermore, a constrained wake model simulation based on the lidar measurements is presented. The measured wind speed deficit in HMFR as well as the measured time series of the meandering

are incorporated into the wake simulations. The incorporation of the wind speed deficit leads to insignificant improvements, which indicates that the shape of the wind speed deficit in the MFR could already be reproduced very well by the recalibrated version. However, only a horizontal line with few scan points has been measured with the lidar system. Thus, a more detailed scan of the wake with a higher temporal resolution might lead to a further decrease of the uncertainties. The incorporation





of the time series of the meandering results in a better agreement with the measured power as well as blade root and tower

bending moments. All in all, the constrained simulations with lidar measurements verify that the conformity between measured

and simulated loads can be enhanced by incorporating the measured meandering as well as the wind speed into the aeroelastic

load simulation.

*Code and data availability.*    Access to lidar and met mast data can be requested by the authors.

*Author contributions.*    IR performed all simulations, post-processed and analyzed the measurement data, and wrote the paper. LS monitored

the load measurements and data acquisition. Furthermore, LS, DS and ND gave technical advice in regular discussions and reviewed the

paper. PD and MB supervised the investigations and reviewed the paper.

*Competing interests.*    The authors declare that they have no conflict of interest.

*Acknowledgements.*    The content of this paper was developed within the project NEW 4.0 (North German Energy Transition 4.0), which is
funded by the Federal Ministry for Economic Affairs and Energy (BMWI).



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
