# Peer review of "Validation of the dynamic wake meandering model with respect to loads and power production"

_Wind Energy Science, 2020_

## Referee Comment (RC1) · Anonymous Referee #1 · 12 Jan 2021

**Review of "Validation of the dynamic wake meandering model with respect to loads and power production" by Inga Reinwardt et al., manuscript number: wes-2020-126**

**1. General comments**

In this article, several versions of the DWM model are compared in terms of power and loads prediction for an onshore wind farm composed of closely spaced machines. Special focus is given to a calibrated version of the DWM model. This article shows the limitations of the Frandsen model compared to the DWM models for reduced spacing between machines. Finally, a lidar assisted version of the DWM model, which includes the wind speed deficit as well as the wake meandering obtained through lidar measurements, is proposed to decrease the uncertainties on the power and loads prediction.

- The article is well written and is pretty clear (see specific and minor comments).
- Try to complete hazy wording (immediate vicinity, considerably more, ...) with more precise data (see specific comments).
- The lidar assisted version of the DWM model presented in this article provides results that are in better agreement with the measurements than those obtained with the "classical" DWM models. It shows therefore that there is still room to improve the physics behind the DWM models and underlines the interest of additional studies focusing on the wake meandering physics. It seems interesting to me to add such a consideration somewhere in the text.

This article shows the influence of the version of the DWM model on power and loads prediction. Through the lidar assisted version of the DWM model, it also highlights the importance of the quality of the wake meandering modeling for power and loads estimation. It has certainly a value for the wind energy research community and I suggest the manuscript for publication after addressing the following comments.

**2. Specific comments**

**Wind farm and measurement equipment**

L 67: "*immediate vicinity*" Please precise by giving the distance of the closest wind farm.

L 79: "recuperator" Do you mean heat exchanger?

L 96: "Considerably more ..." Please precise by giving an order of magnitude.

L 97: "*The measured lidar data are filtered by the power intensity*, ...". This phrasing is confusing considering the sentence at L 92: "*Only measurement results with normal power production are included in the analysis*". Are you actually talking about the "return strength of the laser pulse"? I suggest specifying which power intensity.

L 108-109: "The one-dimensional scan consists of only 11 scan points scanned in a horizontal line..." Please give information about the streamwise positions.

L 109: "... *around 16 s depending on the visibility conditions during the scan.*" What is the order of magnitude of the variation in duration: +- 0.1 s or +- 10 s?

**Load simulation**

L 123-124: *"Furthermore, the multibody model is connected to a controller, which uses the generator speed and the pitch angle from the multibody simulation to calculate the generator torque and the pitch velocity and returns them to the multibody model."* What do you mean by pitch velocity?

L 133-134: *"Auxiliary sensors are added to the turbine model in alaska/Wind to compare the measured loads at the precise position of the strain gauges, the locations of which are given in Section 2."* Please rephrase.

**Dynamic wake meandering model**

Figure 3: I do not understand the arrow going from BEM to small-scale turbulence based on the text at L 162-163: "This small-scale turbulence is calculated with a scaled homogeneous turbulent wind field, which is also generated by a Kaimal spectrum". Could you comment on this, please?

**Lidar assisted load simulation**

L 170-174: *"To clarify that ..., the phrasing Horizontal Meandering Frame of Reference (HMFR) is introduced in Figure 4."* I do not understand the part "*and that the transformed wind speed deficit in the meandering frame of reference still includes the vertical meandering". Please comment on this and rephrase.*

L 191: "... *sigma_y is the standard deviations of the horizontal displacement mu_y."* In Trujillo et al. (2011) and other articles, sigma is the parameter representing the width of the wake. Are you sure that it corresponds to the standard deviation of the horizontal displacement?

L 212-213: *"... and the turbine loads are not increased by an immediate change of the position of the wind speed deficit."* Not clear to me. Could you comment on this, please?

L 214-215: *"The comparison of simulations and measurements shows that the amplitude of the measured time series is more pronounced."* From Fig. 5, it seems to me that it is the amplitude of the simulations (DWM-meas) that is more pronounced than the amplitude of the measured time series. Please clarify.

L 227-228: "... hence the mean wind speed deficit in the HMFR in the DWM model definition should be replaced by the lidar measurements." Not clear to me. You write that "*the measured wind speed deficit shows a coarse distribution at the boundaries of the deficit*", that "*using this coarse curve leads to increased loads in the simulation, which are not feasible"*, and then "*hence the mean wind speed deficit in the HMFR in the DWM model definition should be replaced by the lidar measurements*". Please clarify.

**Results**

L 250-255: *"In summary, the simulated power agrees ... different inflow wind speed at the turbine than the one measured at the met mast and used in the simulations."* How do you justify the discrepancy for 8.25 m/s for which the std (errorbar) is similar to the std (errorbar) of the other wind speeds?

L 260-261: *"Only some slight discrepancies occur between 6m/s and 9m/s, where the simulation overestimates the loads slightly."* Can you give an explanation?

L 262: *"... local maximum around the rated wind speed"* I would say "just below".

L 266-268: *"... the illustration of the measured nacelle wind speed and the met mast prove the fact that the turbine experiences a local momentarily different inflow wind speed and explains the discrepancies."* It seems to me that it is the small number of points in the bin that results in a biased wind speed rather than a local momentarily different inflow wind speed. Could you comment on this?

L 297-298: "*Because of the tilt, the blade faces slightly away from the wind direction during the upward movement ... whereas during the downward movement, the blade faces slightly more towards the wind direction*" Not clear to me. Could you clarify, please?

L 311-314: "*The load is defined in the rotating frame of reference, so that the weight force switches its sign with each rotation, whereas the influence of the aerodynamic force on the edgewise moment does not change the sign. Thus, at one side of the rotor the forces level each other out, while on the other side of the rotor they accumulate.*" The understanding would be eased if you could add a schematic.

L 371: *"... no wind speed deficit is considered in Frandsen's model".* Not clear to me. Could you comment on this?

L 404: *"Another explanation ..."* Which is the first explanation?

L 405-407: *"For the comparison measurements at the closest available lidar range gate that is still outside the rotor area of the downstream turbine is used, thus it happens that the downstream distance used in the simulations is slightly to low."* Not clear to me. Please clarify.

L 408: *"However, the influence should be small, due to the small gradient of the wind speed in downstream direction."* So, it does not justify the differences you observed. Please clarify with L 405-407.

3. **Minor comments**

Table 1: not referenced in the text.

L 182: What is epsilon?

L 241: *"In order to validate the aerodynamic load simulations the following section ..."* → "In order to validate the aerodynamic load simulations, the following section ..."

Figure 14 (a): Please split the legend of the measurements in 2: line for all measurements and circle symbol for 10-min time series for which lidar measurements are available. Remove also gray lines below other symbols.

Figure 14 (b): Add normalized before power for the axis labels and in the legend.

Figures 15 (a), 16 (a) and 17 (a): Idem Fig. 14 (a)

Figure 16 (b): Do not repeat the legend as you didn't for the other figures.

L 395: *"The simulated power over the measured power"* → "The normalized simulated power over the normalized measured power"

L 405: *"For the comparison measurements ..."* → "For the comparison, measurements ..."

L 406: "... is used," → "... are used,"

L 407: "... to low." → "... too low."

---

## Referee Comment (RC2) · Anonymous Referee #2 · 14 Jan 2021

**Review of "Validation of the dynamic wake meandering model with respect to loads and power production" by Inga Reinwardt et al.**

**General comments**

This article is highly relevant to wind farm engineering as it provides a good overview of the capabilities of various modifications of the dynamic wake meandering model.

The text is very well written and the document follows an understandable path. the results are presented in a very clear way, although sometimes the text could be a bit more concise. On the other hand, I understand you want to be complete and present all the findings. In some instances, I would prefer some concrete numbers, instead of brief descriptions of deviations

**Specific comments**

p.2 – l.44-48: is this information really necessary for the reader?

p.5. – l.89: *Measurement results were analyzed from April 2019 to May 2020*. Please rephrase, it sounds like you are worked on analyzing the data in that time span.

P.9. – l.193: What is the induction zone model? Please add a reference or explanation.

P.13. – l.252 ff: You always refer to the rated wind speed, but you do not define it. It can be seen from Figure 7 (a). Nevertheless, I would name it not to create confusion for the reader.

p.16 – l.329: Here you mention the Wöhler coefficient. Is it the m=10 and m=4 specified in the header of the figures? If so please specify it before you use it the first time (Figure 9), otherwise the information in the title distracts the reader.

p.18 – Figure 11: What are the numbers in the graph? You should mention again what they stand for. Maybe also in the caption.

p.22 – l.405-406: *For the comparison measurements at the closest available lidar range gate that is still outside the rotor area of the downstream turbine is used, thus it happens that the downstream distance used in the simulations is slightly to low.* This is hard to understand. Please rephrase

**Technical corrections/comments**

p.4. – l.73: *Whole met mast  as depicted*

p4. – l.82: *three turbines are equipped with load measurement equipment*. They cannot be equipped with measurements but equipment or similar.

p.5: please refer to Table 1 in the text.

P8. – l.169: Firstly instead of First.

P9. – l.191: standard deviation.

P13. – l.250: *shown as  mean values*

P14. – l.283: * and with an ambient*

P15. – Figure 9: The numbers in the graph indicating the number of considered measurements are more confusing then informing. Maybe you could make an extra graph for them, which is valid for all four plots and, which indicates what you say in the text.

P15. – Figure 10: Please be consistent with the fonts and the size of the text that you use for the figures. Figures 10 and 13 look very different from the other plots.

p.22/23 – Figures 14, 15 and 16: Why do you use 3 separate figures here instead of before where you used one graph for all three cases? Furthermore, the blue line in normalized DEL graphs seems not fitting here. As you use the same legend for both graphs (just not in Figure 16?), the reader might get confused by the blue line colour as it would refer to "measurements". So I would suggest to use a different colour for this line.

---

## Author Comment (AC1) · 4 Feb 2021

**Responses to Anonymous Referee #1**

We are very thankful for your valuable comments on the paper. Your comments lead to significant improvements of the paper and have been taken into consideration. We thank you a lot for taking the time to review this paper.

**1. General comments**

- The article is well written and is pretty clear (see specific and minor comments).
- Try to complete hazy wording (immediate vicinity, considerably more, ...) with more precise data (see specific comments).
- The lidar assisted version of the DWM model presented in this article provides results that are in better agreement with the measurements than those obtained with the "classical" DWM models. It shows therefore that there is still room to improve the physics behind the DWM models and underlines the interest of additional studies focusing on the wake meandering physics. It seems interesting to me to add such a consideration somewhere in the text.

Response: The following sentence has been added in the conclusion to address the last bullet point: "*This indicates that there is still room for improvements in the physical description of the meandering, the local turbulence, and the deficit modelling in the DWM model and confirms in particular the significance of further research on wake meandering.*" The remaining comments in the bullet points are directly addressed in the specific comments and minor comments.

**2. Specific comments**

**Wind farm and measurement equipment**

L 67: "*immediate vicinity*" Please precise by giving the distance of the closest wind farm.

Response: Following sentence had been added "*The distance to the next wind farm is approximately 3 km.*"

L 79: "recuperator" Do you mean heat exchanger?

Response: Yes, the wording has been changed to heat exchanger.

L 96: "Considerably more ..." Please precise by giving an order of magnitude.

Response: The following sentence has been added: "In total *around 370 samples could be collected at a turbulence intensity of 12 % in the analysis in Section 7.1.*"

L 97: "*The measured lidar data are filtered by the power intensity, ...*". This phrasing is confusing considering the sentence at L 92: "*Only measurement results with normal power*

*production are included in the analysis*". Are you actually talking about the "return strength of the laser pulse"? I suggest specifying which power intensity.

Response: To clarify what kind of power is meant the sentences in L 97 and L 92 have been changed to: *"Only measurement results, where the turbines operate under normal power production are included in the analysis"* and *"The measured lidar data are filtered by the power intensity from the returned laser beam, which is closely related to the signal-to-noise ratio (SNR) of the measurements."*

L 108-109: "The one-dimensional scan consists of only 11 scan points scanned in a horizontal line..." Please give information about the streamwise positions.

Response: The following sentence has been added: *"Measurements were collected up to a downstream distance of 750 m in 30 m steps"*.

L 109: "... *around 16 s depending on the visibility conditions during the scan.*" What is the order of magnitude of the variation in duration: +- 0.1 s or +- 10 s?

Response: The following phrase has been added: *"The duration of the horizontal line scan is usually around 16 s depending on the visibility conditions during the scan. At poor conditions the scan can take up to 25 s*

**Load simulation**

L 123-124: *"Furthermore, the multibody model is connected to a controller, which uses the generator speed and the pitch angle from the multibody simulation to calculate the generator torque and the pitch velocity and returns them to the multibody model."* What do you mean by pitch velocity?

Response: This refers to the rotational speed of the pitch. The following sentence has been added: *"The pitch velocity refers to the blade angular velocity about the pitch axis during a pitching motion"*

L 133-134: *"Auxiliary sensors are added to the turbine model in alaska/Wind to compare the measured loads at the precise position of the strain gauges, the locations of which are given in Section 2."* Please rephrase.

Response: The sentence has been rephrased to: *"To compare the measured loads with simulations, sensors at the precise position of the strain gauges are added to the turbine model in Alaska/Wind. The locations of the strain gauges are given in Section 2."*

**Dynamic wake meandering model**

Figure 3: I do not understand the arrow going from BEM to small-scale turbulence based on the text at L 162-163: "This small-scale turbulence is calculated with a scaled homogeneous turbulent wind field, which is also generated by a Kaimal spectrum". Could you comment on this, please?

Response: The additional homogenous wind field for the small-scale turbulence is scaled by a factor, which is based on the calculation of the initial deficit, which itself builds on the BEM

theory and the aerodynamics of the turbine. A clarification has been added to the paragraph.

**Lidar assisted load simulation**

L 170-174: *"To clarify that ..., the phrasing Horizontal Meandering Frame of Reference (HMFR) is introduced in Figure 4."* I do not understand the part *"and that the transformed wind speed deficit in the meandering frame of reference still includes the vertical meandering"*. *Please comment on this and rephrase.*

Response: The lidar only scans a horizontal line, hence no two-dimensional Gauss curve can be fitted to the measurement points. Instead, a one-dimensional curve is fitted to the results as outlined in Equation (3). That means that the vertical meandering cannot be captured. The deficit that is transferred into the HMFR is not completely in the MFR because it still includes the vertical meandering. The vertical meandering is less pronounced than the horizontal one, wherefore this simplification only has a small impact. Nevertheless, to make clear that only the horizontal meandering was considered, the abbreviation HMFR is introduced. A more detailed description on how the wind speed deficit in the FFR is transformed to the HMFR is outlined in
Reinwardt, I., Schilling, L., Dalhoff, P., Steudel, D., and Breuer, M.: Dynamic wake meandering model calibration using nacelle-mounted lidar systems, Wind Energy Science, 5, 775–792, https://doi.org/10.5194/wes-5-775-2020, 2020.

The sentence in the paper has been changed to: *To clarify that only the horizontal meandering can be measured and that the transformed wind speed deficit in the MFR is still affected by vertical meandering, because due to the scanning pattern vertical meandering cannot be captured, the phrasing Horizontal Meandering Frame of Reference (HMFR) is introduced in Figure 4*.

L 191: *"... sigma_y is the standard deviations of the horizontal displacement mu_y."* In Trujillo et al. (2011) and other articles, sigma is the parameter representing the width of the wake. Are you sure that it corresponds to the standard deviation of the horizontal displacement?

Response: Yes, that is completely correct. The sentence has been rephrased to: *"...where $A_{1D}$ is a scaling parameter, $\sigma_y$ describes the wind speed deficit width and $\mu_y$ is the horizontal displacement."*

L 212-213: *"... and the turbine loads are not increased by an immediate change of the position of the wind speed deficit."* Not clear to me. Could you comment on this, please?

Response: If the meandering time series wasn't interpolated, the meandering movement would be very sudden and not smooth, because a new position of the wind speed deficit is measured only every 16s (scan time). This is physically incorrect and would lead to increased loads in the simulations, wherefore an interpolation towards smaller time steps has been carried out.

L 214-215: *"The comparison of simulations and measurements shows that the amplitude of the measured time series is more pronounced."* From Fig. 5, it seems to me

that it is the amplitude of the simulations (DWM-meas) that is more pronounced than the amplitude of the measured time series. Please clarify.

Response: The maximum amplitude occurs in the measurements (around 150s). Looking at the power spectrum (Figure 5) it can also be seen that in the low frequency part the energy content from the measurements is higher. Nevertheless, the difference is rather low, so that the sections has been rephrased to: *"The comparison of simulations and measurements shows that the amplitude of the measured time series is slightly more pronounced. Furthermore, at the low frequency part the energy content from the measurements is higher."*

L 227-228: "... hence the mean wind speed deficit in the HMFR in the DWM model definition should be replaced by the lidar measurements." Not clear to me. You write that "*the measured wind speed deficit shows a coarse distribution at the boundaries of the deficit*", that *"using this coarse curve leads to increased loads in the simulation, which are not feasible"*, and then "*hence the mean wind speed deficit in the HMFR in the DWM model definition should be replaced by the lidar measurements*". Please clarify.

Response: The sentence has been rephrased to: *"Using this coarse curve and replacing the wind speed deficit description in the DWM model directly by the measured one leads to increased loads in the simulation, which are not feasible, wherefore the measured wind speed deficit has to be fitted to a smooth curve before applying it in load simulations."*

**Results**

L 250-255: *"In summary, the simulated power agrees ... different inflow wind speed at the turbine than the one measured at the met mast and used in the simulations."* How do you justify the discrepancy for 8.25 m/s for which the std (errorbar) is similar to the std (errorbar) of the other wind speeds?

Response: It can be explained by the momentarily different wind speed at the turbines as explained in the text and confirmed by Figure 8. Also, around 8.25 m/s differences between the met mast and the nacelle anemometer occur and due to the fact that only a few data points could be collected in this area, these outliers significantly influence the overall power curve.

L 260-261: *"Only some slight discrepancies occur between 6m/s and 9m/s, where the simulation overestimates the loads slightly."* Can you give an explanation?

Response: The differences could not be explained by the measured ambient conditions, so that it is assumed that these discrepancies are related to the uncertainty of the load simulation itself. The following sentence has been added to the paper: *"These discrepancies are assumed to be related to the inaccuracy of the load simulation software itself."*

L 262: *"... local maximum around the rated wind speed"* I would say "just below".

Response: Has been changed.

L 266-268: *"... the illustration of the measured nacelle wind speed and the met mast prove the fact that the turbine experiences a local momentarily different inflow wind speed and explains the discrepancies."* It seems to me that it is the small number of points

in the bin that results in a biased wind speed rather than a local momentarily different inflow wind speed. Could you comment on this?

Response: It could be the case that the measured wind speed is biased but we think that the comparison of met mast and nacelle anemometer measurements in Figure 8 proves the assumption of a locally different inflow. Furthermore, as you already mentioned in the previous comments, the standard deviation around 8.25m/s is not much higher than in the lower wind speed bins, which is in contradiction to the biased wind speed assumption. Nevertheless, we cannot exclude the fact that a higher bias influences the results, so that the following changes has been added: *"The differences in the edgewise moment and the power around the rated wind speed as well as the illustration of the measured nacelle wind speed and the met mast supports the hypothesis that the turbine experiences a local momentarily different inflow wind speed and explains the discrepancies. Furthermore, due to the low amount of data points in this region, the measured wind speed might be biased."*

L 297-298: *"Because of the tilt, the blade faces slightly away from the wind direction during the upward movement ... whereas during the downward movement, the blade faces slightly more towards the wind direction"* Not clear to me. Could you clarify, please?

Response:  To provide a better explanation of the phenomenon, the following graphical illustration was added to the appendix of the paper.

[Figure]

**Figure A1.** Schematic illustration of the flapwise blade root bending moment according to Reinwardt (2017). The aerodynamic force $F_{r,z}$, the gravitational force $F_{g,z}$ as well as the total force $F_z$ perpendicular to the rotor plane at different wake situations are marked.

The explanation of the different pronounced maxima is related to the aerodynamic force and the rotor tilt. The aerodynamic forces on a blade segment are a function of the apparent wind velocity, which is a vector composed of the motion of the blade and the incoming wind. Due to the turbine tilt, the apparent wind velocity is slightly lower during the upward movement, and the aerodynamic force is reduced. The blade faces slightly away from the wind direction during the upward movement, whereas during the downward movement, the blade faces slightly more towards the wind direction, which results in an increase of the aerodynamic force. The aerodynamic force (the green arrow) is higher outside the wake on the left side of the figure than the aerodynamic force on the right side of the figure outside the wake.  Similarly, the aerodynamic force in the wake is higher on the right side. At wake conditions, the increase is stronger when the wind speed deficit coincides with the upward movement of the rotor (left side of the figure), so that a higher alternating load at the blade occurs and the maximum is more pronounced in comparison to the case, where the wind speed coincides with the downwind movement of the rotor (right side of the figure).

L 311-314: *"The load is defined in the rotating frame of reference, so that the weight force switches its sign with each rotation, whereas the influence of the aerodynamic force on the edgewise moment does not change the sign. Thus, at one side of the rotor the forces level each other out, while on the other side of the rotor they accumulate."*
The understanding would be eased if you could add a schematic.

Response:  To provide a better explanation of the phenomenon, the following graphical illustration of the edgewise moment was added to the appendix of the paper.

[Figure]

**Figure A2.** Schematic illustration of the edgewise blade root bending moment according to Reinwardt (2017). The aerodynamic force $F_{r,y}$, the gravitational force $F_{g,y}$ as well as the total force $F_y$ in the rotor plane at different wake situations are marked.

The explanation of the curve of the edgewise moment over the wind direction in wake conditions is also related to the influence of the aerodynamic force, which has an influence on the edgewise moment due to the tilt of the rotor and the gravitational force. The edgewise load is defined in the rotating frame of reference, so that the weight force switches its sign with each rotation (see direction of red arrows in the figure inside and outside of the wake), whereas the influence of the aerodynamic force on the edgewise moment does not change the sign (green arrows). Thus, at one side of the rotor the forces level each other out, while on the other side of the rotor they accumulate. If the deficit is on the side where the forces level each other out, the alternating load increases in comparison to a situation without wake, whereas when the wind speed deficit is on the side where both aerodynamic and weight force are facing in the same direction, the alternating load is decreased.

L 371: *"... no wind speed deficit is considered in Frandsen's model"*. Not clear to me. Could you comment on this?

Response: The Frandsen model only describes a wake added turbulence without any wind speed deficit. It is a common way in industry to not consider any wind speed deficit in the Frandsen model and to run the simulations with ambient wind speed. This approach has been chosen here as well. To clarify that the added wake turbulence is calculated, the name Frandsen model is replaced by Frandsen wake added turbulence model.

L 404: *"Another explanation ..."* Which is the first explanation?

Response: The sentence has been changed to: *"An explanation for the differences and uncertainties can be found in the different downstream distance, which is used in the simulations."*

L 405-407: *"For the comparison measurements at the closest available lidar range gate that is still outside the rotor area of the downstream turbine is used, thus it happens that the downstream distance used in the simulations is slightly to low."* Not clear to me. Please clarify.

Response: The lidar system does not measure directly at the downstream position of the rotor because only fixed 30 m range gates are measured, so that it has been measured some meters ahead of the turbine. To clarify the following sentence has been added: *"The lidar specifically measures in 30 m range gates, so that no measurements are available at the exact position of the downstream turbine."*

L 408: *"However, the influence should be small, due to the small gradient of the wind speed in downstream direction."* So, it does not justify the differences you observed. Please clarify with L 405-407.

Response: It does not justify all the differences. It leads to a slightly inaccurate description of the wake. Furthermore, it should only lead to a too pronounced deficit, but looking at the power deficit in Figure 14, sometimes the power is also overestimated, which can't be explained by this. To clarify this, the following adjustments were made in the paper: *"However, the influence should be rather small due to the small gradient of the wind speed in downstream*

*direction, hence it does not explain all differences completely. Especially, an overestimation of the power cannot be explained by the too low downstream distance, wherefore it is assumed that some discrepancies are related to a bias in the determination of the ambient conditions and/or the load simulation itself."*

**3. Minor comments**

Table 1: not referenced in the text. Response: A reference has been added.

L 182: What is epsilon? Response: Epsilon is the radial distance from the hub normalized by the radial position where the induction is half of the centreline induction. The formula to calculate epsilon is now given in the paper.

L 241: *"In order to validate the aerodynamic load simulations the following section ..."* → "In order to validate the aerodynamic load simulations, the following section ..." Response: Has been adjusted.

Figure 14 (a): Please split the legend of the measurements in 2: line for all measurements and circle symbol for 10-min time series for which lidar measurements are available. Remove also gray lines below other symbols. Response: Has been adjusted.

Figure 14 (b): Add normalized before power for the axis labels and in the legend. Response: Has been adjusted.

Figures 15 (a), 16 (a) and 17 (a): Idem Fig. 14 (a)  Response: Has been adjusted.

Figure 16 (b): Do not repeat the legend as you didn't for the other figures. Response: Has been adjusted.

L 395: "*The simulated power over the measured power*" → "The normalized simulated power over the normalized measured power" Response: Has been adjusted.

L 405: *"For the comparison measurements ..."* à "For the comparison, measurements ..." Response: Has been adjusted.

L 406: "... is used," → "... are used," Response: Has been adjusted.

L 407: "... to low." → "... too low." Response: Has been adjusted

---

## Author Comment (AC2) · 4 Feb 2021

**Responses to Anonymous Referee #2**

We are very thankful for your valuable comments on the paper. Your comments lead to significant improvements of the paper and have been taken into consideration. We thank you a lot for taking the time to review this paper.

**Specific comments**

p.2 – l.44-48: is this information really necessary for the reader?

Response: These references are less relevant for the paper than the previous ones and can be neglected.

p.5. – l.89: *Measurement results were analyzed from April 2019 to May 2020.* Please rephrase, it sounds like you are worked on analyzing the data in that time span.

Response: The sentence has been rephrased to: *"Measurement results from April 2019 to May 2020 have been used in the analysis."*

P.9. – l.193: What is the induction zone model? Please add a reference or explanation.

Response: It is explained above in Equation (2). A reference is given there as well.

P.13. – l.252 ff: You always refer to the rated wind speed, but you do not define it.
It can be seen from Figure 7 (a). Nevertheless, I would name it not to create confusion for the reader.

Response: Rated wind speed is 11 m/s and added in text.

p.16 – l.329: Here you mention the Wöhler coefficient. Is it the m=10 and m=4 specified in the header of the figures? If so please specify it before you use it the first time (Figure 9), otherwise the information in the title distracts the reader.

Response: The Wöhler coefficient is given in the title. A hint is given when Figure 9 is described.

p.18 – Figure 11: What are the numbers in the graph? You should mention again what they stand for. Maybe also in the caption.

Response: The Wöhler coefficient is given in the title. A hint is given when Figure 9 is described.

p.18 – Figure 11: What are the numbers in the graph? You should mention again what they stand for. Maybe also in the caption.

Response: The numbers illustrates the measured 10-min time series per wind direction bin. But I changed it and added a secondary axes with a bar graph according to the hints in the Technical corrections/comments.

p.22 – l.405-406: *For the comparison measurements at the closest available lidar range gate that is still outside the rotor area of the downstream turbine is used, thus it happens that the downstream distance used in the simulations is slightly to low.* This is hard to understand. Please rephrase

Response: The lidar system does not measure directly at the downstream position of the rotor because only fixed 30 m range gates are measured, so that it has been measured some meters ahead of the turbine. To clarify this, the following sentence has been added: "*The lidar specifically measures in 30 m range gates, so that no measurements are available at the exact position of the downstream turbine.*"

**Technical corrections/comments**

p.4. – l.73: *Whole met mast and as depicted* Response: Has been adjusted.

p4. – l.82: *three turbines are equipped with load measurements equipment*. They cannot be equipped with measurements but equipment or similar. Response: Has been adjusted to: "*At last, at three turbines load measurement equipment is installed.*"

p.5: please refer to Table 1 in the text. Response: Has been adjusted.

P8. – l.169: Firstly instead of First. Response: Has been adjusted.

P9. – l.191: standard deviation*s*. Response: Has been reformulated.

P13. – l.250: *shown as a mean values* Response: Has been reformulated.

P14. – l.283: *with and and with an ambient* Response: Has been reformulated.

P15. – Figure 9: The numbers in the graph indicating the number of considered measurements are more confusing then informing. Maybe you could make an extra graph for them, which is valid for all four plots and, which indicates what you say in the text.
Response: To improve the graph's readability, a secondary axes with a bar plot has been added. It provides the number of measurements for each point in the graph.

P15. – Figure 10: Please be consistent with the fonts and the size of the text that you use for the figures. Figures 10 and 13 look very different from the other plots. Response: Has been adjusted.

p.22/23 – Figures 14, 15 and 16: Why do you use 3 separate figures here instead of before where you used one graph for all three cases? Furthermore, the blue line in normalized DEL graphs seems not fitting here. As you use the same legend for both graphs (just not in Figure 16?), the reader might get confused by the blue line colour as it would refer to "measurements". So I would suggest to use a different colour for this line.

Response: The graphs need to be bigger, hence they don't fit in one graph. Otherwise it is hard to read them with all the marks inside. The legend is adjusted in every graph. The blue line

with error bars are the complete measurements, whereas the blue circles are only the measurements were lidar data are available.